# The COVID-19 Pandemic: Early Ripple Effects in Pediatric Palliative Care

**DOI:** 10.3390/children9050642

**Published:** 2022-04-29

**Authors:** Linda Marisol Bustamante, Regina Okhuysen-Cawley, Julia Downing, Stephen R. Connor, Mary Ann Muckaden, Marianne Phillips, Andrea Icaza, Nicole Garzon, Yuriko Nakashima, Kelsi Morgan, David Mauser, Michelle Grunauer

**Affiliations:** 1Unidad Nacional de Oncología Pediátrica, Guatemala 01011, Guatemala; mbustamantederuiz@gmail.com; 2Divisions of Pediatric Critical Care and Palliative Medicine, Baylor College of Medicine and Texas Children’s Hospital, Houston, TX 77030, USA; 3International Children’s Palliative Care Network, Bristol BS1 1BU, UK; julia.downing@icpcn.org; 4Worldwide Hospice Palliative Care Alliance, London WC1X 9JG, UK; sconnor@thewhpca.org; 5Department of Palliative Medicine, Tata Memorial Centre, Kharghar, Navi Mumbai 410210, India; muckadenma@tmc.gov.in; 6Departments of Oncology and Palliative Medicine, Perth Children’s Hospital, Nedlands, WA 6009, Australia; marianne.phillips@health.wa.gov.au; 7School of Medicine, Universidad San Francisco de Quito (USFQ), Quito EC170157, Ecuador; aicazaf@usfq.edu.ec (A.I.); mgrunauer@usfq.edu.ec (M.G.); 8Posgrado de Pediatría UIDE, Hospital Axxis, Quito EC170157, Ecuador; ngarzon@asig.com.ec; 9Department of Pediatrics, Universidad de Guadalajara, Guadalajara 44100, Mexico; ynakashima@hcg.gob.mx; 10Department of Pediatrics, Baylor College of Medicine, Houston, TX 77030, USA; kxmorga1@texaschildrens.org; 11Division of Palliative Care (Pediatric Advance Care Team), Baylor College of Medicine, Houston, TX 77030, USA; dbmauser@texaschildrens.org

**Keywords:** pediatric palliative care, COVID-19, SARS-CoV-2, global, pediatric, burnout, resilience

## Abstract

Palliative care, which aims to provide comprehensive, interdisciplinary, holistic care to children, adolescents and adults with life-threatening, and ultimately life-limiting conditions, is a discipline that has emerged as an integral component of healthcare systems throughout the world. Although the value of life-affirming palliative care (PC) has been shown across many domains, funding and acceptance of palliative care teams have been variable: some hospital systems have free-standing, dedicated interdisciplinary teams while, in many instances, palliative care services are provided “pro bono” by individuals with a special interest in the discipline, who provide PC in addition to other responsibilities. In this article, we hope to highlight some of the observations on the early effects of the COVID–19 pandemic on the provision of PC in children.

## 1. The Emergence of the COVID-19 Pandemic

Initial reports of a novel coronavirus causing severe disease emerging from Wuhan, a large industrial city in China, came to the attention of the World Health Organization (WHO) in December 2019 [1]. Situation reports, designed to highlight the impact of this emerging pathogen, began circulating on 21 January 2020, emphasizing the need for a vigorous containment and mitigation effort to prevent the dangerous outbreak from transforming into a pandemic [1]. Two hundred and eighty-two cases of the novel coronavirus-associated illness had been confirmed in four countries on that day (China, Japan, Republic of Korea and Thailand) [1]. World Health Organization situation reports issued in the following days chronicled the rapid spread of the pathogen, prompting a declaration of a “global health emergency” on 30 January 2020, when close to 8000 cases had been confirmed in 18 countries throughout Asia, Australia, Europe and North America, reflecting the shocking ease of transmission through global air travel. At that time, 170 deaths due to the infection had been reported in China alone [2].

By 11 March 2020, on the day *WHO’s Situation Report 51* was published, the virus had been identified in 114 countries. Around 118,319 cases of clinical infection had been diagnosed, and 4292 people had died of novel coronavirus-associated disease, causing the WHO to upgrade the global health emergency to pandemic status, the first ever to be caused by a coronavirus [3]. That landmark Situation Report included a paragraph on collaboration by the International Federation of the Red Cross, UNICEF and WHO to mitigate transmission in schools and other institutions attended by children [3].

As of late September 2021, more than one year later, it was evident that a parallel pandemic of inaction and misinformation, which began the day the global health emergency was first declared, had enabled the rapid progression of this catastrophic pandemic. According to the website maintained by the Johns Hopkins Coronavirus Resource Center, as of 27 September 2021, approximately 500,000 new cases were reported globally each day during what appeared to be the decline of the third global wave since the pandemic was initially declared. Unfortunately, over 426 million cases have been reported; and almost 6 million deaths have occurred due to COVID-19 as of 23 February 2022 [4].

It is estimated that well over a million children have been orphaned, or lost a custodial relative, with all the attendant risks and complications this represents for children worldwide [5]. Unfortunately, the vaccines developed at warp speed to mitigate the severity of disease and ultimately bring the pandemic under control are not uniformly available, highlighting the plight of the most vulnerable citizens of the world, as is exemplified by what is happening in Africa [5].

While many countries are seeing “plateaus” after severe, prolonged spikes, and mortality, per se, appears to be decreasing to some degree in some countries as clinicians become more proficient in managing the acute infection—particularly when patients develop respiratory failure—it is now apparent that in many cases, severe, prolonged, and possibly permanent disability will hamper physical, emotional, and financial recovery [6]. The limited access to effective vaccines has allowed a rapid proliferation of pathogenic variants in highly vulnerable parts of the world, as demonstrated by the highly transmissible and virulent Delta variant [6] before it was displaced, a few months later, by the exquisitely transmissible Omicron variant.

## 2. The Impact of the COVID-19 Pandemic on Children throughout the World

The ongoing, prolonged COVID-19 pandemic has had tremendous repercussions for children and adolescents throughout the world, as demonstrated by the UNICEF report published a little over a year after the pandemic began [7]. A significant decline in important indicators of overall health, safety, well-being and cognitive development has been observed, particularly occurring in children who have been orphaned or bereft of supportive family figures. The loss of nurturing family members, in turn, has compounded surging poverty and vulnerability.

Sadly, there has been an increase in violence against children and intimate partners inside the home, increased violence in society at large, and a spike in forced marriages in adolescence, all of which have contributed to a sharply increased incidence of social isolation, depression, substance abuse, and the other mental health crises that accompany grief.

Furthermore, there is evidence of impaired academic progress due to school closures, especially in the poorest urban areas. Additionally, those living in remote parts of the world lack the internet access needed for online educational programs [8,9,10,11,12]. Tragically, these and many other detrimental effects are expected to be enduring [7].

Economic repercussions have been severe, particularly amongst the poorest individuals, who have had limited, if any, opportunities to prepare for this contingency and a lack of government financial support [13]. Many countries have had to mandate prolonged or repeated shutdowns, with the subsequent financial repercussions due to “downsizing”: reduction in salaries and benefits packages, even in relatively stable economies; decreased affordability of child daycare, and a decrease or loss of disposable income crucial to self-employed workers who subsist on day-to-day income and lack healthcare and other benefits [14]. Many of these individuals saw their ability to earn a living practically evaporate from one day to the next as shutdowns were implemented, and in many cases, at very short notice [14].

## 3. Impact of the COVID-19 Pandemic on Healthcare Systems

Hospital systems throughout the world—particularly those in World Bank low and middle income (LMIC) economies—were suddenly faced with budgetary constraints forcing reductions in staff, reallocations of already scarce physical space and other resources, and the dissolution of clinical teams judged to be less critical to institutional missions [15,16]. Concerns over pathogen transmission through aerosols have mandated the use of safety protocols, such as the use of social distancing, personal protective equipment (PPE) and special isolation measures within hospitals that were found to be effective in other pandemics [13]. Many healthcare institutions throughout the world have experienced a sudden, severe scarcity of PPE, forcing these institutions to limit consulting services to conserve PPE for essential frontline workers, in the face of a fragile supply chain [17]. Healthcare teams have become physically and mentally exhausted with extended shifts under precarious conditions and emotionally overwhelmed in a setting dominated by death [18]. The world was shocked, just a few months ago, by the plight of patients and families in India, for example, where the lack of oxygen and oxygen supply-related accidents undoubtedly contributed to the mortality as the highly transmissible highly pathogenic Delta variant emerged [19].

Hospital system operating expenses quickly exceeded “pre-COVID” budgets, and essential medicines became unavailable, inaccessible, and unaffordable, particularly in chronically underfunded LMICs, although there are recent reports of shortages in high-income countries as well [20]. Sadly, corruption has also complicated the pandemic and has significantly contributed to the climate of uncertainty.

## 4. The COVID-19 Pandemic: Impact on Palliative Care and Other Healthcare Providers

The ongoing pandemic has created severe constraints for many palliative care teams throughout the world. Many teams had to lay off staff due to funding shortfalls, as donations were canceled or redirected to acute care and intensive care services at the onset of the pandemic [13]. This is tragic because many of the 21 million children with serious illnesses who would benefit from an integrated, embedded level of at least a primary level of palliative support [21] stopped receiving palliative care services altogether. It must be remembered that the one thing pandemics cannot do is halt the progression of other illnesses. In other words, the “illness clock” never pauses—it simply marches on—or even *accelerates* as access to routine healthcare services for vaccine-preventable diseases of childhood, becomes more difficult even for healthy children. It is also important to note that it is known that significant underlying chronic illnesses are risk factors for severe and fatal COVID-19 infection [15].

The rapid progression of the pandemic in many countries was accompanied by a steep increase in patient mortality, a phenomenon few professionals had witnessed previously. Moreover, the use of personal protective equipment (PPE) caused isolation from human contact between clinicians and patients, and between team members accustomed to the warmth and close interaction so characteristic of palliative care [13]. The scarcity of PPE forced scores of clinicians, particularly in LMICs, to ignore nature’s cues for self-care in favor of safeguarding PPE supplies [22].

Healthcare workers worry about the risks of bringing the pathogen home to a beloved family member, and, particularly during the peaks of the pandemic, have had to worry whether the patient on the next gurney might be someone they know or love [19,20]. Clinicians remain at an elevated risk for worry, burnout, and the very real fear of infecting a loved one—particularly when the loved one is a family member or even a patient at high risk for severe disease [17,23]. Regrettably, while most healthcare professionals were recognized by their communities, cheered on their way to work, or tired as they were, on their way home, some were injured or even killed in incidents of “violence, harassment or stigmatization” as noted in the journal *Lancet* on 5 September 2020, when the pandemic was just ramping up into its second global peak [24].

Although fewer children and young adults have become critically ill as a result of COVID-19 in comparison to older individuals, especially the elderly adults with underlying comorbidities, serious, life-threatening illness has occurred in some children, adolescents and young adults as a result of the disease. In some cases, 25% of new recorded infections requiring hospitalization are occurring in children. According to one recent report, among adolescents hospitalized for COVID-19, almost one-third have required ICU admission [25]. Among children and adolescents admitted to the ICU for COVID-19, some patients have required extensive intervention to manage complications of COVID-19, including mechanical ventilation, extracorporeal membrane oxygenation (ECMO), and even lung transplant.

This highlights how this infection has life-limiting implications even in countries prosperous enough to offer advanced modes of respiratory support, including cadaveric lung transplants in the more seriously affected patients. [26,27]. Children with underlying illnesses have been particularly vulnerable to fatal COVID pneumonia. Moreover, COVID-associated multisystem inflammatory syndrome in children (MIS-C), which can lead to life-threatening complications, including acute heart failure, has been increasingly recognized [28]. Pregnant young adults can have catastrophic complications, making public health measures, including timely immunization, of paramount importance [17,23].

The challenges posed by exposure to large numbers of dying patients, as has been observed throughout the world, and by the losses suffered personally by so many clinicians (thousands of nurses, doctors, pharmacists, social workers and chaplains working in hospital settings have become gravely ill and died of COVID-19) have made education on the personal and professional tasks of grieving, and the practical and educational aspects of bereavement increasingly recognized as the pandemic rages on. On average, every death of a grandparent, parent, sibling, spouse or child from COVID-19 could leave approximately nine people bereaved [29].

## 5. Finding “Silver Linings” in the Midst of a Global Catastrophe

Fortunately, palliative care services—including pediatric teams—are comprised of highly dedicated, compassionate individuals who are resilient and hard-wired to help the suffering. Palliative care teams—even pediatric teams—have been able to pivot to provide a supportive presence to all healthcare workers, particularly in hospital settings where so many adults have been dying alone despite the heroic efforts of their clinical teams [15]. This has been an unimaginably stressful experience for frontline clinicians witnessing several patients dying during a single shift.

It is evident that the pandemic has brought out a sense of humility in clinicians wary of palliative care colleagues and practices, making it easier for palliative care providers to gain the trust of other medical professionals. Our “other” medical colleagues, previously so wary and even suspicious, have come to know us—to finally sit down with us, and to carefully listen to us for the very first time [30]. They began to regard us, as we deserve, as the clinical service that innovates and values interdisciplinary teamwork, that communicates carefully and teaches communication skills. They began to see that we care for our own and embrace patients and families so that they can continue to navigate this great uncertainty with a steady step, one day at a time.

Some hospital-based palliative care clinicians, which were previously regarded as belonging to secondary “filler services”, are now viewed and respected as indispensable colleagues, able to mitigate suffering in a large number of seriously ill and dying patients of all ages [15]. With patience and humility, palliative care professionals have earned not only trust but have also been afforded the opportunity to teach “*other*” teams the essential instruments in our “toolkit”—particularly the essential tools of communication, teamwork, care of dying patients and their families, especially when symptom control is challenging at the end of life [15]. The pandemic has also raised self-awareness regarding woefully inadequate self-care in healthcare workers [14,22,31].

Some teams, while in lockdown, were able to reimagine, innovate and implement solutions necessary to continue supporting children and families with ongoing palliative care needs [32]. Families and teams need us—and we knew that we had to keep working. Almost every medical publication has devoted space to the COVID-19 disease, including clinical and social repercussions as the pandemic unfolds. Although clinicians, primarily in LMIC countries, have utilized technology with “end-to-end” encryption applications such as WhatsApp to support seriously ill patients and families during terminal illness and bereavement, many hospital systems rapidly implemented telemedicine programs that have made it possible for families to continue to receive interdisciplinary palliative care remotely [9].

These telemedicine options had to “ramp up” very quickly, particularly in communities with a rapid spread of the pathogen and emerging variants [13]. It is anticipated that many of the internet-based telemedicine resources will continue to play an important part in the delivery of care across all specialities, including palliative care, after the global emergency posed by the pandemic starts to abate.

Palliative care professionals have seized the opportunity to support our own adult and pediatric colleagues on the frontlines, living the most difficult professional experience one could ever imagine. We have also found novel ways of supporting our very own. At first, pediatric palliative care members of various social media platforms commiserated. It quickly became apparent that a more structured approach to professional distress and suffering was needed. This was the impetus behind a new initiative by the Pediatric Commission of the Latin American Palliative Care Association (ALCP). A “check-in” with members, still early in the pandemic, revealed the depth of the distress felt by the group. Although only a few clinicians “zoomed in” to the initial informal “check-in” organized by the pediatric section of the ALCP, almost 200 clinicians from the Americas, from Canada to Chile have tuned in to the next few webinars [33,34,35,36].

The International Children’s Palliative Care Network (ICPCN) started preparing and disseminating briefing notes early in this global disaster. The website, maintained by the organization, contains educational resources translated into 12 languages that continue to serve clinicians worldwide, as before the pandemic. Several expert-led webinars have complemented state-of-the-art materials published by the organization over the last few months [33,34,35,36]. Previous years’ “Hats on for Children’s Palliative Care” was themed to reflect the ongoing pandemic on 8 October 2021—Hats 1 was “Hats and Masks on for Children’s Palliative Care 2021” (see Figure 1).

Other professional associations have also stepped up to examine the implications of the pandemic using the lens of bioethics. Discussions about crisis resource allocation and other ethical dilemmas in the era of COVID-19 have become more common in all parts of the world, from the special circumstances surrounding intensive care unit bed assignments to the need, in some cases, to withhold or withdraw life-sustaining medical interventions [27]. Palliative care practitioners have ample experience in complex medical decision-making and communication of difficult information. They have as such, been able to from the early days of the pandemic, to facilitate discussions, deliberations, partnered medical decision-making, resolution of ethical dilemmas, and the design of protocols for advanced symptom control and care at the end of life, under the challenging, isolating conditions unique to each region. [30].

Despite the challenges imposed by “lockdown conditions”, palliative care providers have found novel opportunities in the unusual challenge of this particular and deadly pandemic. With expertise in addressing physical, psychosocial, and spiritual needs during times of human crisis, palliative care teams have much to contribute to helping other health care workers improve human connectedness and supporting patients and families to deal with stress and fear [30]. We have had to quickly find ways to share and teach the vital tool of communication that is the essence of our work to teams unaccustomed to sharing or receiving difficult news [25,30]. It was heartening to see how some teams have learned quickly, and this important skill has trickled down, so that healthcare students in all disciplines now have this important tool, regardless of their future professional roles and goals.

Interestingly, some hospitals, where spiritual support was unavailable, actually recruited and trained ordained priests so that they could safely, in full PPE, minister to the gravely ill and the teams who cared for them, with daily prayer sessions and Catholic mass, and other rituals.

Finally, the pandemic has forced us to recognize the importance of self-care: we are coming to a much better understanding of the neurobiology of burnout, and factors favoring resiliency [37]; we are recognizing the importance of awareness of our own physiologic stress responses, and the importance of incorporating mindfulness into our daily personal and professional lives. Individual and team resiliency strategies will doubtless help us to not only *survive*, but also *thrive* during this global health emergency, and other catastrophes that will follow.

## 6. Future Directions

Advocating for children has always been one of the most important—and most rewarding—public health measures. Our innate predisposition to care for others has helped us embrace—despite the significant personal hardships we have all experienced—the challenges posed by this historic pandemic. We have turned these challenges into opportunities.

It is uncertain *whether*, *how* and *when* palliative care teams dispersed by the pandemic will reorganize, and if so, what institutional resources will be available. It is safe to assume that access to pediatric palliative care will be even more challenging once the pandemic resolves. We have come to see, indeed, how countries are traversing the public health uncertainty sparked by the pandemic in very different ways.

Hopefully, global immunization programs that include children, along with practices that promote safe and equitable access to education and combat misinformation will help to bring the pandemic under control, or at least, promote the “new normal” concept. It is hoped that the reorganization of temporarily dispersed teams and the development of pediatric palliative care services continue as they had before the pandemic.

## Figures and Tables

**Figure 1 children-09-00642-f001:**
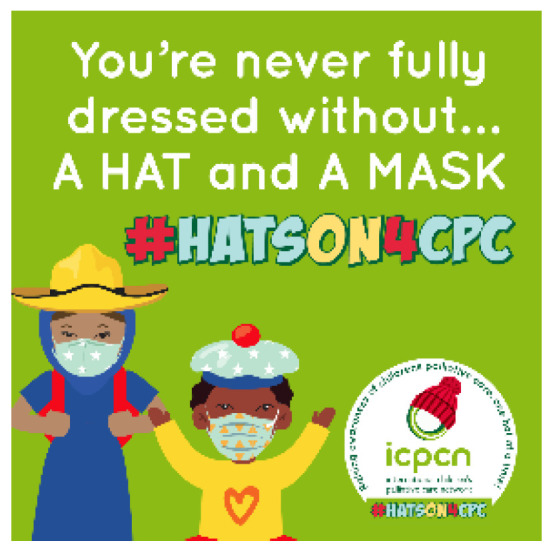
Hats and Masks on for Children’s Palliative Care 2021.

## Data Availability

This manuscript describes the effects of the pandemic in diverse communities utilizing clinician narratives, media reports, medical literature and public domain statements.

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
