# Peer review of "The COVID-19 Pandemic: Early Ripple Effects in Pediatric Palliative Care"

_children, 2022, doi:10.3390/children9050642_

Round 1

Reviewer 1 Report

very interesting review article - i think overall it could be shortened and tightened up.

this paragraph could be reworded and softened - it reads as overly judgemental.

It is evident that the pandemic has brought out a sense of humility in clinicians wary 178 of palliative care colleagues and practices, making it easier for palliative care providers to 179 gain the trust of other medical professionals. Our “other” medical colleagues, previously 180 so wary and even suspicious, have come to know us – to finally sit down with us, and to 181 carefully listen to us for the very first time [32]. They began to regard us, as we deserve, 182 as the clinical service that innovates, that values interdisciplinary teamwork, that com- 183 municates carefully and teaches communication skills. They began to see that we care for 184 our own and embrace patients and families so that they can continue to navigate this great 185 uncertainty with a steady step, a day at a time. 

Author Response

Thank you very much for taking the time to review our manuscript, and for your valuable observations.

We have revised the manuscript in its entirety, including the paragraph on interactions with other hospital services, which seemed judgmental.  One of the major accomplishments we have noticed, practicing in a variety of inpatient and outpatient clinic settings, is that other clinical services once wary of our role in the care of hospitalized children, and even our adult colleagues, with whom we did not share any clinical responsibilities, recognized the importance of impeccable communication and symptom control.

We hope we have been able to positively influence the care of patients of all ages throughout the pandemic, once it resolves into a more “endemic” stage, and hopefully, the care of all seriously-ill patients, regardless of their expected disease trajectory.

Reviewer 2 Report

Dear Authors,

Thank you for collaborating on this important topic.  The idea on the content are original and interesting.  However, I found the text difficult to read.  I suggest dividing the text into sections with sub titles such as "The emergence of COVID-19", "Pandemic effect on healthcare"  "Pandemic effect on palliative care "  "Pandemic effect on pediatric palliative care" and "Future Directions"  or "Implications", etc.

I would also suggest adding a table to highlight your main findings of how the COVID-19 pandemic has impacted pediatric palliative care.

In an effort to focus the reader's attention, you may also consider significantly shortening or deleting most of the first three paragraphs. 

Stylistically, it is confusing that the piece is written in third person and in first person.  I suggest being consistent with one or the other; traditionally the third person is used in scientific writing.  An example of the first person would be line 242.

Also, there is liberal use of third party editorialization, such as in line 244 "it was heartening to see..." which should be addressed.  Unless, of course, this is intended to be an editorial, in which case that should be made clear to the reader.

Author Response

Thank you very much for taking the time to review our manuscript, and for your very valuable suggestions.

The manuscript was extensively revised and was updated to reflect the continuing effects of the pandemic.

Sadly, COVID-19 continues to evolve with variants that continue to devastate families and healthcare systems.

You will find that we have incorporated subtitles to improve the overall flow of the manuscript, as recommended. Creation of tables will be easier once we know a little bit more about the pandemic as it transitions – hopefully soon – into an endemic pattern.